# Diffusion-weighted magnetic resonance sequence and CA125/CEA ratio can be used as add-on tools to ultrasound for the differentiation of ovarian from non-ovarian pelvic masses

Patrick Nunes Pereira[1,2], Sophie Françoise Derchain[1], Adriana Yoshida[1]*,
Ricardo Hoelz de Oliveira Barros[2], Rodrigo Menezes Jales[1], Luís Otávio Sarian[1]

1 Faculty of Medical Sciences, Department of Obstetrics and Gynecology, State University of Campinas—Unicamp, Campinas, São Paulo, Brazil, 2 Section of Imaging, Sumaré State Hospital, State University of Campinas, Sumaré, São Paulo, Brazil

* adriana122013@gmail.com

## Abstract

### Objective

To provide a straightforward approach to the sequential use of ultrasound (US), magnetic resonance (MR) and serum biomarkers in order to differentiate the origin of pelvic masses, making the most efficient use of these diagnostic resources.

### Study design

This is a cross-sectional study with 159 patients (133 with ovarian and 26 with non-ovarian tumors) who underwent surgery/biopsy for an adnexal mass. Preoperative CA125 and CEA serum measurements were obtained and a pelvic/abdominal ultrasound was performed. Preoperative pelvic MR studies were performed for all patients. Morphological and advanced MR sequences were obtained. Using a recursive partitioning algorithm to predict tumor origin, we devised a roadmap to determine the probability of non-ovarian origin using only statistically significant US, laboratory and MR parameters.

### Results

Upfront US classification as ovarian versus non-ovarian and CA125/CEA ratio were significantly associated with non-ovarian tumors. Signal diffusion (absent/low *versus* high) was the only MR parameter significantly associated with non-ovarian tumors. When upfront US designated a tumor as being of ovarian origin, further MR signal diffusion and CA125/CEA ratio were corrected nearly all US errors: patients with MR signal diffusion low/absent and those with signal high but CA125/CEA ratio ≥25 had an extremely low chance (<1%) of being of non-ovarian origin. However, for women whose ovarian tumors were incorrectly

**Data Availability Statement:** All relevant data are within the paper and its Supporting Information files.

**Funding:** author: SFD number 2012/15059-8 São Paulo Research Foundation, Fapesp https://fapesp.br the funders had no role in study design, data collection and analysis, decision to publish, or preparation of the manuscript.

**Competing interests:** The authors have declared that no competing interests exist.

rendered as non-ovarian by upfront US, neither MR nor CA125/CEA ratio were able to determine tumor origin precisely.

## Conclusion

MR signal diffusion is an extremely useful MR parameter to help determine adnexal mass origin when US and laboratory findings are inconclusive.

## Introduction

The preoperative definition of tumor origin (ovarian vs. non-ovarian) is key to the treatment planning of women with adnexal masses [1,2]. Preoperative investigation protocols for suspected malignant non-ovarian pelvic masses should encompass exams for the gastrointestinal tract (e.g. colonoscopy), and the surgical approach planning for these cases often requires the presence of medical teams capable of performing gastrointestinal or urologic interventions. Women with a suspected malignant non-ovarian pelvic mass should undergo extensive extra-pelvic investigation, in order to ascertain the tumors' primary site and disease extent [3].

Unfortunately, clinical manifestations, imaging and laboratory findings can be elusive and lead to incorrect identification of tumor origin in many cases [4]. It is not uncommon that unexpected intraoperative difficulties arise, such as the necessity of gastrointestinal and extra-pelvic interventions without the patient being prepared to undergo such interventions; or the surgical team not being properly trained to deal with non-gynecological patients. A Cochrane Systematic Review underscored the concept that women with gynecologic malignancies have a longer survival when treated in specialized gynecologic oncology centers, compared to patients treated elsewhere (general or community hospitals). Patients with malignant ovarian tumors incorrectly identified as non-ovarian masses could be incorrectly referred to such non-specialized centers and therefore receive suboptimal treatment [5].

The rarity of cases and the dearth of data on large cohorts of women with ovarian/non-ovarian pelvic masses are obstacles for the development of diagnostic protocols that discern women who should be treated by a gynecologic oncology surgeon [2,6]. Many attempts, however, are being made to tackle the problem using currently available diagnostic tools. The ultrasound (US) and cancer antigen 125 (CA125)-based malignancy prediction model of the International Ovarian Tumor Analysis (IOTA) group called ADNEX has been designed to categorize adnexal masses as benign, borderline, malignant (initial, advanced) or metastatic [7].

Our study aimed to provide a straightforward approach to the sequential use of US, magnetic resonance imaging (MRI), and serum biomarkers in order to adequately differentiate the origin of pelvic masses making the most efficient use of these diagnostic resources.

## Methods

This prospective study was conducted at the Women's Hospital of the University of Campinas, a tertiary cancer center. The study was approved by the University Research Ethics Committee (protocol #1092/2009 and #008/2010) and included patients afferent to the gynecological oncology department from October 2011 to September 2020. Follow-up for this study lasted through March 2022.

Sample size for this study was estimated taking into consideration the difference in probability of ovarian malignancy in patients grouped according to the general US classification proposed by Timmerman et al. [8]. Other parameters of interest for sample size calculation were

alpha error = 10%, beta error = 80%. With these parameters in mind, it was estimated that at least 32 patients in each US category (malignant/non-malignant) should be included. The actual numbers of women in this study, according to the proposed simplified US criteria [8], were: ovarian malignant = 47 (31.7%), other = 112 (68.3%), totaling 159 patients.

We non-consecutively included women referred to our hospital due to an adnexal or pelvic mass, without prior biopsy and treatment. Women were excluded if they did not undergo surgery or biopsy and were discharged from Hospital or were submitted to follow-up, or died before further investigation. After signing the informed consent form, patients were submitted to a physical exam; blood samples were then collected for serum marker measurement. An US evaluation of the pelvis was scheduled for all women. After performing US, 278 cases were scheduled for MRI, which were performed in the Sumaré State Hospital, an affiliated hospital. When indicated, diagnostic and/or treatment surgical procedures were performed. The indication of surgery was based on the pelvic exam, preoperative biomarker levels, US (IOTA simple rules) [8] and MRI results, without knowledge of Ovarian-Adnexal Reporting Data System Magnetic Resonance Imaging (O-RADS MRI) score. Symptomatic women with image exams suggestive of benign tumors, or those with suspicious pelvic masses had indication of surgery or biopsy. Of the 278 women initially enrolled, we excluded 117 that did not have a histopathological diagnosis and 2 because of missing CEA results. After exclusions, data from 159 women was available for analysis (Fig 1).

The final histological diagnosis was established following the World Health Organization International Classification of Ovarian Tumors guidelines [9] by a team of pathologists specialized in gynecological cancer.

Biomarkers assessment, US, surgery and histopathological analysis were performed at this single institution (Women's Hospital). More than one ovarian tumor was found in 15 women; each adnexal tumor was described separately in US and MRI reports. Only the tumor with the worst prognosis was considered for statistical analysis.

Histopathological specimens were obtained from either surgery (143 cases), percutaneous biopsy (13 cases), Pipelle® biopsy (2 cases), or through colonoscopy (1 case). All US and MRI assessments were performed before the definition of the histological diagnosis.

## Ultrasonography

Ultrasound examinations were performed by a level 3 examiner in gynecological ultrasound, or were carried out under the supervision of experienced examiners at the Ultrasound Technical Section of the main Hospital. The ultrasound machines used were Voluson Expert 730 (GE Healthcare Ultrasound, Milwaukee, WI, USA) and Toshiba Xsario SSA– 660A (Toshiba Medical Systems Corporation, Japan). Ultrasound evaluation was performed initially via a transabdominal approach with ~~the patient~~ full bladder ~~full~~; next, a transvaginal scan with empty bladder was performed. Adnexal masses were described following the IOTA's terms and definitions (Simple Rules) [8,10]. Also, Color Doppler was used (utilizing pulse repetition frequency 0,3–0,6 kHz) and a color score was attributed to each adnexal mass. For some cases, the examiner gave a subjective assessment of whether the mass was expected to be an ovarian or non-ovarian/indeterminate tumor. Sonographers performed all the exams without awareness of histological diagnosis and clinical data.

## Magnetic Resonance Imaging (MRI)

One radiologist specialized in pelvic, and another radiologist with expertise in upper abdomen MRI (both with >10 years of practice) have independently evaluated the MRI scans. Both radiologists had no prior knowledge of clinical information, biomarker results, patients' previous

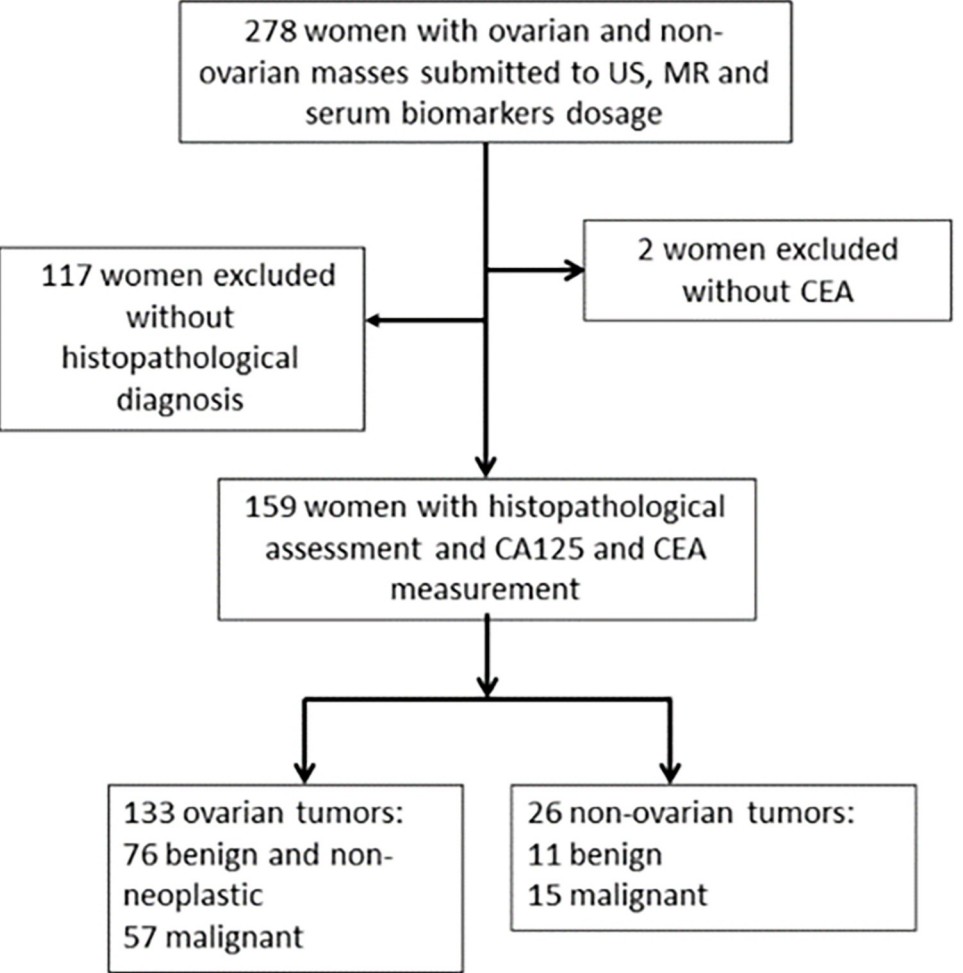

**Fig 1. Flowchart describing patient inclusion and exclusion in the study.**

imaging or histological diagnoses. Scans were acquired on a General Electric 1.5 Tesla machine (GE Signa HDxt®) using a pelvic phased-array coil.

We used a protocol aimed at assessing adnexal masses, which consisted of T2-weighted multiplanar sequences (axial, sagittal and coronal), a T1-weighted sequence in and out phase, diffusion-weighted sequence (b = 0, 500 and 1000) and T1-weighted sequences, with fat sat, before and after intravenous contrast injection with a power injection at a rate of 3.5mL/sec. The post-dynamic study consisted of 5 sequential acquisitions, with an interval of 30 seconds between them, with each sequence having an acquisition time varying between 10 and 13 seconds. The beginning of the first sequence was 21 seconds after the injection of intravenous contrast. An additional upper abdomen diffusion-weighted sequence was performed for screening of distant metastasis (solid organs or lymphadenopathy). We used an echo planar diffusion-weighted sequence (b = 0, 500 and 1000 s/mm2). The T2-weighted signal intensity within the solid component of the adnexal mass ~~is~~ was compared to urine within the bladder. A visual/qualitative analysis was performed: we defined diffusion-weighted high signal intensity in the solid portion when signal in lesion ~~is~~ was greater than urine, when b = 1000 s/mm2. We emphasize that always in this analysis, we used the ADC (apparent diffusion coefficient) map as a reference, in order to avoid the effect T2 shine through *(pseudo restriction)*.

The following MRI parameters were evaluated: size, septa [no, single, two or more], septa thickness [thin, thick], T2-weighted signal intensity within solid tissue [absent/low, medium/high], b = 1000 s/mm2 –weighted signal intensity within solid tissue [absent/low, medium/high], wall enhancement [yes, no], time-signal intensity curve within solid tissue [type 1/type 2, type 3], ascites [yes, no], peritoneal implants [yes, no] and metastasis [yes, no]).

### Biomarker measurement

Blood samples collected from patients were stored in serum separator tubes. They were left to clot for at least 30 minutes before centrifugation. Blood samples were centrifuged 1300g for 10 min, and serum was separated into aliquots and stored at -80˚C until analysis of biomarkers.

**Cancer antigen 125 (CA125) and carcinoembryonic antigen (CEA).** Serum CA125 and CEA were determined by the CA125 II and CEA tests, respectively. Both biomarkers were measured in serum samples through chemiluminescence (Cobas e411, Roche Diagnostics GmbH, Mannheim, Germany). CA125 levels were expressed in U/ml, and CEA in ng/ml.

### Statistical analysis

Data were analyzed using the R Environment for Statistical Computing Software [11]. Statistical calculations were performed within 95% confidence intervals (CIs), considering P<0.05 as significant, except for the conditional inference tree, for which a 90% (p<0.10) confidence interval was assumed. Ovarian tumors were classified into malignant, benign, borderline, non-neoplastic and metastatic groups; non-ovarian tumors were classified into malignant or benign non-ovarian tumors according to histopathologic diagnosis. Clinical characteristics of groups classified histologically as ovarian or non-ovarian were compared using the chi-square test for categorical variables and Kruskal-Wallis test for continuous variables.

We assessed whether CA125/CEA ratio and MRI parameters (tumor size, septa [single, two or more], septa thickness [thin, thick], T2-weighted signal intensity within solid tissue [absent/low, medium/high], b = 1000 s/mm2 –weighted signal intensity within solid tissue [absent/low, medium/high], wall enhancement [yes, no], time-signal intensity curve within solid tissue [type 1/type 2, type 3], ascites [yes, no], peritoneal implants [yes, no], metastasis [yes, no]) were associated with non-ovarian origin using bivariate tests as appropriate (Chi-squares, Fisher's test). Next, we fitted a logistic regression model for the discrimination of ovarian from non-ovarian tumors using as explanatory variables the upfront ultrasound evaluation impression (ovarian/non-ovarian), CA125/CEA ratio and MRI parameters as listed above. The explanatory variables significantly associated with non-ovarian status (upfront ultrasound impression, CA125/CEA ratio, and MRI signal diffusion) were then included in a recursive partitioning regression model [12], from which a conditional inference tree was generated. The global null hypothesis of independence between input variables (explanatory variables) and the response (non-ovarian/ ovarian mass) was tested. Branches of the generated inference tree bifurcate when a statistically significant association was detected (P<0.05) (Fig 2).

### Results

Table 1 shows the key clinical characteristics of women with ovarian and non-ovarian tumors. A total of 159 women were included in the study, 26 (16.4%) of whom had non-ovarian tumors. Age, menopausal status and CA125 levels were not statistically related to tumor origin, whereas women with non-ovarian tumors had significantly higher CEA levels (55.1 vs 5.8 ng/mL, p = 0.03) and significantly lower CA125/CEA ratios (38.3 vs. 333.5; p = 0.006).

In Table 2, the final pathological diagnoses are listed.

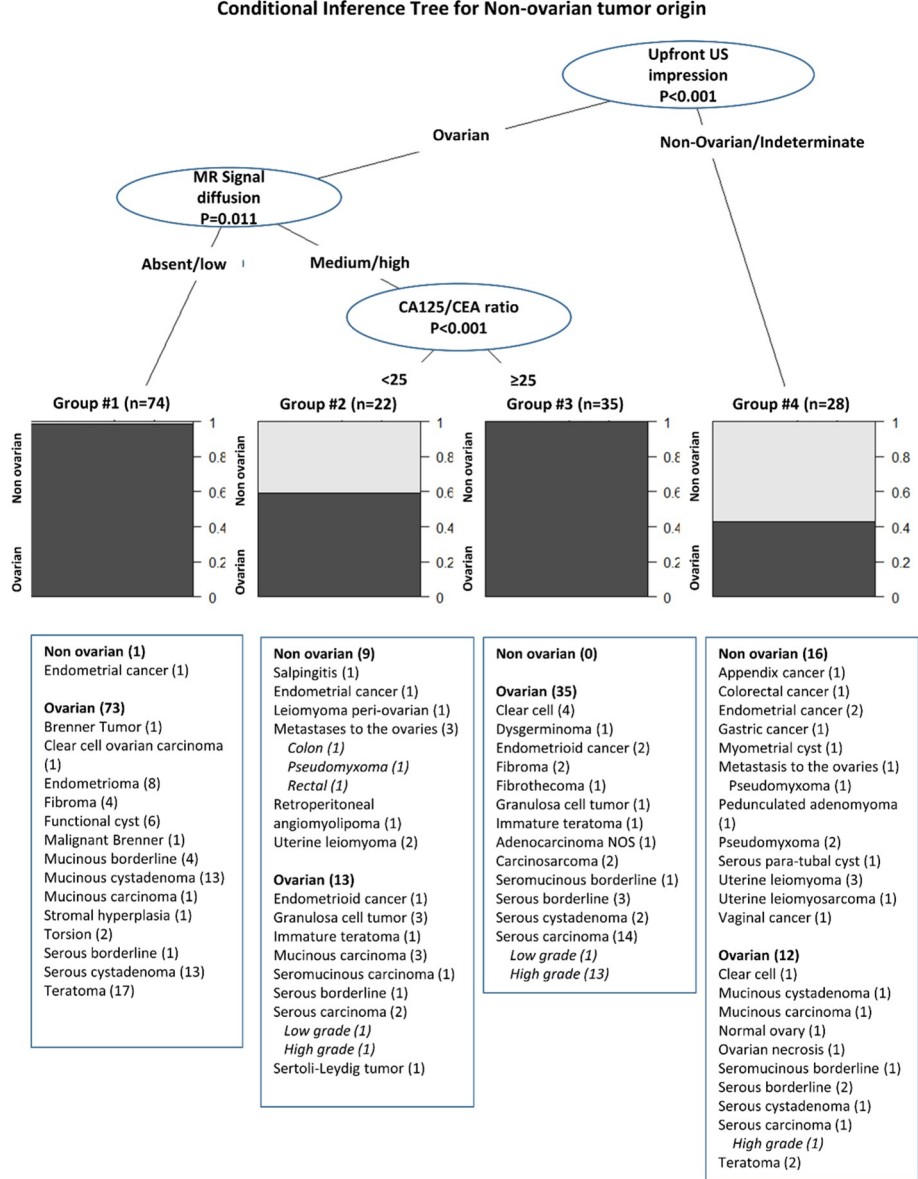

**Fig 2. Conditional Inference Tree (CIT) for non-adnexal origin of the tumor in the adnexa.** Each group denotes a subset of women with the characteristics indicated in the CIT. Significance level set to *alpha* = 95% (tree will branch only where p<0.05). Variables included in CIT modeling: Upfront ultrasound classification of tumors into ovarian/non-ovarian, CA125/CEA ratio, MRI signal in diffusion-weighted.

**Table 1. Key features of women with adnexal masses.**

| | Final diagnosis | | | | |
|---|---|---|---|---|---|
| | Ovarian (n = 133) | | Non-ovarian (n = 26) | | p |
| Age (mean (sd)) | 51.6 | (17.6) | 52.8 | (13.6) | 0.74 |
| Postmenopausal (n (%)) | 67 | (50.4%) | 16 | (61.5%) | 0.40 |
| CA 125 levels in U/mL (mean (sd)) | 506.1 | (1590.7) | 125.0 | (314.7) | 0.25 |
| CEA levels in ng/mL (mean (sd)) | 5.8 | (36.2) | 55.1 | (171.6) | **0.03** |
| CA125/CEA ratio (mean (sd)) | 333.5 | (1130.8) | 38.3 | (86.1) | **0.006** |

**Table 2. Histopathological diagnosis of 159 study cases.**

| Histopathological diagnosis | n (%) |
|---|---|
| **Ovarian** | 133 (83.6%) |
| **Non ovarian**<br>(*including metastasis to the ovaries*) | 26 (16.4%) |
| **Ovarian benign** | 57/159 (35.8%) |
| Epithelial | 31 |
| Serous | 16 |
| Mucinous | 14 |
| Brenner tumor | 1 |
| Teratoma | 19 |
| Fibroma | 6 |
| Fibrothecoma | 1 |
| **Non neoplastic ovarian** | 19/159 (11.9%) |
| Endometrioma | 8 |
| Functional cyst | 6 |
| Ovarian torsion | 2 |
| Normal ovary | 1 |
| Ovarian stromal hyperplasia | 1 |
| Ovarian necrosis | 1 |
| **Non ovarian benign** | 11/159 (6.9%) |
| Uterine leiomyoma | 5 |
| Myometrial cyst | 1 |
| Peri ovarian leiomyoma | 1 |
| Chronic salpingitis | 1 |
| Serous paratubal cyst | 1 |
| Pedunculated adenomyoma | 1 |
| Retroperitoneal angiomyolipoma | 1 |
| **Epithelial Ovarian Cancer** | 36/159 (22.6%) |
| Serous | 17 |
| Seromucinous | 1 |
| Mucinous | 5 |
| Endometrioid | 3 |
| Clear cells | 6 |
| Brenner | 1 |
| Carcinosarcoma | 2 |
| Adenocarcinoma not otherwise specified | 1 |
| **Borderline ovarian tumor** | 13/159 (8.2%) |
| Serous | 7 |
| Mucinous | 4 |
| Seromucinous | 2 |
| **Rare ovarian malignant** | 8/159 (5%) |
| Sertoli-Leydig | 1 |
| Dysgerminoma | 1 |
| Granulosa cell tumor | 4 |
| Immature teratoma | 2 |
| **Metastasis to the ovaries** | 4/159 (2.5%) |
| Colorectal cancer | 1 |
| Rectosigmoid cancer | 1 |

(*Continued*)

**Table 2.** (Continued)

| Histopathological diagnosis | n (%) |
|---|---|
| Pseudomyxoma peritonei | 2 |
| **Other primary sites, malignant** | 11/159 (6.9%) |
| Pseudomyxoma peritonei (ovaries not compromised) | 2 |
| Uterine leiomyosarcoma | 1 |
| Gastric cancer | 1 |
| Appendix cancer | 1 |
| Colorectal cancer | 1 |
| Endometrial cancer | 4 |
| Vaginal cancer | 1 |

Table 3 shows a comparison of ovarian vs. non-ovarian tumors as related to the MRI parameters tumor size, septum, septum thickness, T2-weighted signal intensity within solid tissue, b = 1000 sec/mm2 –weighted signal intensity within solid tissue, wall enhancement, time-signal intensity curve within solid tumors, presence of ascites, peritoneal implants and metastases. Of these, septum thickness (p = 0.007) and wall enhancement (p = 0.02) were positively associated with non-ovarian etiology.

As described in statistical analysis, we produced a multivariate logistic regression model for the diagnosis of non-ovarian etiology, taking as explanatory variables the upfront ultrasound evaluation impression (ovarian/non-ovarian), CA125/CEA ratio and MRI parameters listed in Table 3. Table 4 shows the explanatory variables with the lowest p-values for the association with non-ovarian final status, as derived from the logistic regression model. The effect sizes and statistical significances of these associations are listed in Table 4.

**Fig 2** conveys a brief pathological description of the tumors allocated to each group (ovarian vs. non-ovarian). Upfront US incorrectly classified 22 ovarian masses: 10 out of 96 cases classified as "ovarian" by upfront US were found to be of non-ovarian origin (these cases are listed in groups #1 and #2) and 12 out of 28 cases classified as "non-ovarian" by upfront US were actually of ovarian origin (these cases are listed in group #4). It seems clear from the inferential tree that, when upfront US designates a tumor as ovarian, further MRI signal diffusion and CA125/CEA ratio are capable of nearly offsetting US error: patients with MRI signal diffusion low/absent and those with signal high but CA125/CEA ratio $\geq$25 had an extremely low probability ($<$1%) of being of non-ovarian origin. Group #2 encompasses those patients whose tumors were classified as ovarian by upfront US and that had medium/high MRI diffusion signal and CA125/CEA ratio $<$25. This group encapsulates nearly all non-ovarian tumors erroneously classified as ovarian by US: salpingitis, endometrial cancer, leiomyoma, a retroperitoneal lipoma and 3 metastatic tumors. Unfortunately, for women whose ovarian tumors were incorrectly classified by upfront US as non-ovarian, neither MRI nor CA125/CEA ratio was able to correctly determine tumor origin (group #4). Additional figures are shown in Supporting Information S1–S5 Figs.

## Discussion

Our study shows the usefulness of MRI signal intensity in diffusion-weighed sequence, in association with CA 125 and CEA levels, as adjuvant tools for the correct discrimination of complex pelvic mass cases as of ovarian or non-ovarian origin. The interpretation of our conditional inference tree led us to conclude that, for women with an US impression of an adnexal mass of ovarian origin, a medium/high signal diffusion combined with CA125/CEA

**Table 3. Comparison of MRI parameters between ovarian and non-ovarian tumors.**

| MRI parameters | Ovarian (n = 133) | Non-Ovarian (n = 26) | p-value |
|---|---|---|---|
| *Size (cm)* | 10.4±6.25 | 10.2±4.65 | 0.88 |
| *Septum* | | | |
| single | 11(15.9%) | 1(8.3%) | |
| two or more | 58(84.1%) | 11(91.7%) | 0.68 |
| *Septum thickness* | | | |
| thin | 26(37.7%) | 0 | |
| thick | 43(62.3%) | 12(100%) | 0.007 |
| *T2-weighted signal intensity within solid tissue* | | | |
| low | 13(19.1%) | 3(15.8%) | |
| medium/high | 55(80.9%) | 16(84.2%) | 1 |
| *b = 1000 sec/mm2 –weighted signal intensity within solid tissue* | | | |
| low | 13(19.4%) | 2(10.5%) | |
| medium/high | 54(80.6%) | 17(89.5%) | 0.50 |
| *Wall enhancement* | | | |
| no | 23(26.7%) | 0 | |
| yes | 63(73.3%) | 16(100%) | 0.02 |
| *Time–signal intensity curve within solid tissue* | | | |
| type 1 | 13(20.6%) | 4(23.5%) | |
| type 2 | 31(49.2%) | 6(35.3%) | |
| type 3 | 19(30.2%) | 7(41.2%) | 0.60 |
| *Ascites* | | | |
| No | 64(48.1%) | 12(46.2%) | |
| Yes | 69(51.9%) | 14(53.9%) | 1 |
| *Peritoneal implants* | | | |
| No | 103(83.7%) | 19(73.1%) | |
| Yes | 20(16.3%) | 7(26.9%) | 0.26 |
| *Metastasis* | | | |
| No | 118(96.7%) | 25(96.2%) | |
| Yes | 4(3.3%) | 1(3.8%) | 1 |

The denominators used to calculate the percentages vary according to the available data.

ratio ≥25 indicated a probability of 100% of diagnosing ovarian tumors; the majority of which (85.7%) being malignant. By contrast, after upfront US suggesting an ovarian origin for the adnexal mass, low or absent MRI signal diffusion was strongly associated with benign and non-neoplastic ovarian lesions. In addition, an US suggesting ovarian origin, followed a

**Table 4. Significant factors associated with non-adnexal tumors after multivariate logistic regression model adjustment.**

| Factors | Adjusted odds ratio (95%CI) | P (Wald test) |
|---|---|---|
| Upfront US (ovarian vs. non-ovarian) | 26.3 (7.7 to 89.9) | <0.001 |
| CA125/CEA ratio (<25 vs. ≥25) | 0.24 (0.07 to 0.79) | 0.01 |
| | | |
| *Magnetic Resonance Parameters* | | |
| Signal diffusion (absent/low vs. high) | 0.26 (0.08 to 0.8) | 0.019 |

medium/high MRI signal diffusion and then a CA125/CEA ratio <25 identified 59% of primary ovarian tumors (all of them malignant), which indicates that a medium/high MRI signal was more relevant than the biomarkers ratio in the determination of tumor origin. These findings are of utmost clinical relevance, since proper treatment strategies can be tailored for patients sharing these features.

Our findings are in alignment with the literature, since the vast majority of adnexal tumors that show high signal in MRI diffusion-weighted sequences are malignant [13,14]. In the present study, we chose to apply the MRI morphological characteristics obtained in our previous study [14] and added the use of CA125/CEA ratio, which involves simple biomarkers which are widely available in the preoperative setting and bear a known relationship with tumor origin (whether ovarian epithelial or gastrointestinal). It is important to note that our study pioneered an attempt to integrate MRI findings with serum biomarkers in order to define the primary site of adnexal masses.

In the biomarkers realm, Sagi-Dain and cols. [15] also reported an association between CA125/CEA ratio ≥25 and ovarian tumors, which in turn is in perfect alignment with our own findings. Biomarkers can be useful tools in the preoperative prediction of malignancy in women with suspicious adnexal masses, ranging from isolated CA125 to some algorithms based on a combination of multiple biomarkers, e.g., risk of ovarian malignancy algorithm (ROMA), OVA1®, Overa® [16]. However, there is a dearth of data about the use of the available biomarkers in the discrimination of ovarian from non-ovarian tumors. Sorensen et al. in a study with 355 malignant ovarian and non-ovarian tumors demonstrated that when CA125/CEA ratio >25, 82% of the ovarian cancers were correctly identified [17]. A study involving 495 patients showed that CA125 associated with CEA did not perform better than CA125 alone in the discrimination of benign from malignant adnexal masses. However, it is worth noting that CA125/CEA ratio ≥25 was significantly associated with primary and non-metastatic ovarian tumors, relative risk (RR) = 2.4, 95% CI, 1.3–4.6 [15].

In the context of preoperative evaluation of suspicious adnexal masses, US is the triage imaging to characterize an adnexal mass, and MRI is indicated to indeterminate masses on US, especially those large pelvic masses with extension to the upper abdomen [18]. However, since the most common non-gynecological tumor types metastasizing to the ovaries are breast, colorectal, gastric, and appendix tumors, additional exams such as colonoscopy, upper abdomen computed tomography, mammography, and upper gastrointestinal endoscopy may be necessary in the preoperative scenario [3].

One strong point of the study was that the confirmation of all pelvic mass nature was given by histology, but we must highlight some limitations of our study. In order to reflect the day-to-day clinical practice, we chose to evaluate a wide range of (non) ovarian adnexal tumors. However, endometriomas and germ cell tumors are easily identifiable using US and/or MRI. Consequently, one may argue that there is no point in applying this algorithm for typical tumors. Nevertheless, our algorithm can provide valuable help in doubtful cases, such as very large adnexal lesions (when anatomical landmarks are lost) and in cases of malignant tumors, in which early and correct referral directly impacts patients' survival [5]. For most pelvic masses it is possible to determine the primary site (ovarian vs extraovarian) using basic and problem-solving MRI sequences, associated with indirect and/or direct MRI imaging findings such as the identification of the ovarian vein drainage of the lesion that indicates an ovarian origin. By contrast, when both ovaries are discernible the possibility of ovarian origin of the mass is ruled-out [19,20].

In synthesis, this study may present useful data for radiologists and gynecologic surgeons concerned about the origin of adnexal/pelvic masses in subsets of their patients with difficult to interpret US findings. To be prepared to treat a gastrointestinal or other non-ovarian

condition is always preferable to discovering the true nature of a pelvic mass in the intraoperative period, since technical resources may not be readily available. Using a straightforward, relatively simple strategy of combining information from MRI and serum biomarkers, the extra resources needed to treat women with non-ovarian masses correctly can be summoned before treatment starts.

## Supporting information

**S1 Fig. Adnexal mass in a 48 year-old postmenopausal woman, with CA125 51.8 U/ml and CEA 0.69 ng/ml.** Axial T2 -weighted spin-echo image reveals a right adnexal mass, predominantly solid (green arrow).
(TIF)

**S2 Fig. Granulosa cell tumor.** Axial B-1000 diffusion weighted demonstrated areas of high signal in solid tissue in the right adnexal mass (green arrow). CA125/CEA ratio = 75.07 and in the Conditional Inference Tree this tumor was located in group 3 (probability of 100% for ovarian tumor).
(TIF)

**S3 Fig. Serous paratubal cyst.** Axial T2 weighted spin-echo image shows an adnexal cystic mass (red arrows), with no septa or solid portion, adhered to the right ovary (blue arrows). It was positioned in group 4 of Conditional Inference Tree.
(TIF)

**S4 Fig. A 41-year-old premenopausal woman, with CA125 10.84 U/ml, CEA 1.45 ng/ml and CA125/CEA ratio 7.47.** Axial T2 -weighted spin-echo image reveals a left adnexal mass, predominantly solid (red arrows).
(TIF)

**S5 Fig. Retroperitoneal angiomyolipoma.** Axial B-1000 diffusion weighted demonstrated multiple areas of high signal in the left adnexal mass (green arrows). Based on conditional inference tree, this mass was categorized in group 2 (probability of 40% for ovarian tumor).
(TIF)

## Author Contributions

**Conceptualization:** Sophie Françoise Derchain, Adriana Yoshida.

**Data curation:** Patrick Nunes Pereira, Adriana Yoshida, Ricardo Hoelz de Oliveira Barros, Rodrigo Menezes Jales.

**Formal analysis:** Luís Otávio Sarian.

**Funding acquisition:** Sophie Françoise Derchain.

**Investigation:** Patrick Nunes Pereira, Adriana Yoshida, Ricardo Hoelz de Oliveira Barros, Rodrigo Menezes Jales.

**Methodology:** Patrick Nunes Pereira, Adriana Yoshida.

**Project administration:** Sophie Françoise Derchain.

**Supervision:** Sophie Françoise Derchain, Luís Otávio Sarian.

**Writing – original draft:** Patrick Nunes Pereira, Adriana Yoshida, Ricardo Hoelz de Oliveira Barros, Rodrigo Menezes Jales, Luís Otávio Sarian.

**Writing – review & editing:** Sophie Françoise Derchain, Adriana Yoshida, Luís Otávio Sarian.

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
