## [Decision Letter · Decision Letter 0]

28 Sep 2022

PONE-D-22-12006Diffusion-weighted Magnetic Resonance sequence and CA125/CEA ratio can be used as add-on tools to ultrasound for the differentiation of ovarian from non-ovarian pelvic massesPLOS ONE

Dear Dr. Yoshida,

Thank you for submitting your manuscript to PLOS ONE. After careful consideration, we feel that it has merit but does not fully meet PLOS ONE’s publication criteria as it currently stands. Therefore, we invite you to submit a revised version of the manuscript that addresses the points raised during the review process.

We look forward to receiving your revised manuscript.

Kind regards,

Federico Ferrari, MD, PhD

Academic Editor

PLOS ONE

2.We note that you have stated that you will provide repository information for your data at acceptance. Should your manuscript be accepted for publication, we will hold it until you provide the relevant accession numbers or DOIs necessary to access your data. If you wish to make changes to your Data Availability statement, please describe these changes in your cover letter and we will update your Data Availability statement to reflect the information you provide.

Reviewers' comments:

Reviewer's Responses to Questions

**Comments to the Author**

1. Is the manuscript technically sound, and do the data support the conclusions?

Reviewer #1: Yes

Reviewer #2: Yes

2. Has the statistical analysis been performed appropriately and rigorously? 

Reviewer #1: Yes

Reviewer #2: Yes

3. Have the authors made all data underlying the findings in their manuscript fully available?

Reviewer #1: Yes

Reviewer #2: No

4. Is the manuscript presented in an intelligible fashion and written in standard English?

Reviewer #1: Yes

Reviewer #2: No

5. Review Comments to the Author

Reviewer #1: The topic is very interesting, the association of MRI with the CA125/CEA ratio is unprecedented in the literature and the study is of great clinical importance. I present some suggestions and questions below.

Introduction, lines 6-63: “In contrast, patients with malignant ovarian tumors incorrectly identified as non-ovarian masses could have been better managed by gynecologic oncology surgeons [5].” I suggest that the authors describe studies that demonstrate that the prognosis of ovarian cancer is better when it is initially managed by a gynecologic oncology surgeon. The authors could also describe this aspect in the discussion, which is related to the clinical importance of the topic studied.

Methods, lines167-168: “Ovarian tumors were classified into malignant, benign and borderline, non-neoplastic and metastatic groups;”. In the metastatic group, I did not understand whether secondary ovarian tumors were included. In table 2, I understood that the authors included ovarian metastasis in the “non ovarian” group. I suggest this be explained further.

Results: The data in lines 221 – 224 are exactly the same as the data shown in table 4. This repetition should be avoided.

Reviewer #2: The authors studied 159 patients with ultrasound, CA125 and CEA biomarkers and MRI and built a CIT tree for non-adnexal origin of the tumor in the 191 adnexa. The results showed diffusion weighted MRI signal can be helpful in determining adnexal mass origin when ultrasound and CA125 and CEA ratio are not conclusive. This study is interesting though more patients from multi-centers are still required to validate this method. One main question is the threshold (line 147-152 on page 8) to define the low and high signal in all the MRI images are not clarified, that could make the conclusion questionable. The scanning protocols and data should be made available. Since histology is available for all the patients, an ROC analysis for the different imaging methods and combination with molecular biomarkers are suggested.

6. PLOS authors have the option to publish the peer review history of their article (what does this mean?). If published, this will include your full peer review and any attached files.

Reviewer #1: No

Reviewer #2: No

---

## [Author Response · Author response to Decision Letter 0]

9 Nov 2022

Reviewer #1

Comment 1: “The topic is very interesting, the association of MRI with the CA125/CEA ratio is unprecedented in the literature and the study is of great clinical importance. I present some suggestions and questions below.”

Response 1: Thank you very much for the revision of our manuscript and for the encouraging comments. We hope your concerns have now been tackled in this revised version of our manuscript. 

Comment 2: “Introduction, lines 6-63: “In contrast, patients with malignant ovarian tumors incorrectly identified as non-ovarian masses could have been better managed by gynecologic oncology surgeons.” I suggest that the authors describe studies that demonstrate that the prognosis of ovarian cancer is better when it is initially managed by a gynecologic oncology surgeon. The authors could also describe this aspect in the discussion, which is related to the clinical importance of the topic studied.”

Response 2: This is a very thoughtful comment, indeed. We have now added the following text to the introduction: “A Cochrane Systematic Review underscored the concept that women with gynecologic malignancies have a longer survival when treated in specialized gynecologic oncology centers, compared to patients treated elsewhere (general or community hospitals). Patients with malignant ovarian tumors incorrectly identified as non-ovarian masses could be incorrectly referred to such non-specialized centers and therefore receive suboptimal treatment.”

Comment 3: “Methods, lines 167-168: “Ovarian tumors were classified into malignant, benign and borderline, non-neoplastic and metastatic groups;”. In the metastatic group, I did not understand whether secondary ovarian tumors were included. In table 2, I understood that the authors included ovarian metastasis in the “non ovarian” group. I suggest this be explained further.”

Response 3: Thank you for pointing this out. Indeed, in the metastatic group, we included 4 cases of metastasis to the ovaries. In fact, we believe that a misunderstanding could have emerged from the designation of these cases in the original version of the manuscript (they were described only as “ovarian metastases”). In this revised version, we have fixed this by using the term “Metastasis to the ovaries”. In addition, also to ensure clarity in Table 2, where it read “Extra ovarian malignant”, we rephrased as “Other primary sites, malignant”. We hope these modifications can make Table 2 less confusing to the readers. 

Comment 4: “Results: The data in lines 221 – 224 are exactly the same as the data shown in table 4. This repetition should be avoided.”

Response 4: Thank you for pointing this out. We removed the redundancies. 

 

Reviewer #2: 

Comment 1: “The authors studied 159 patients with ultrasound, CA125 and CEA biomarkers and MRI and built a CIT tree for non-adnexal origin of the tumor in the 191 adnexa. The results showed diffusion weighted MRI signals can be helpful in determining adnexal mass origin when ultrasound and CA125 and CEA ratio are not conclusive.”

Response 1: Thank you very much for the time and effort put into reviewing our manuscript. We are particularly thankful for your encouraging words. We have now carefully addressed each of your concerns. We hope this revised manuscript can now be deemed acceptable for publication. 

Comment 2: “One main question is the threshold (line 147-152 on page 8) to define the low and high signal in all the MRI images are not clarified, that could make the conclusion questionable. The scanning protocols and data should be made available.”

Response 2: Very good point. The scanning protocols are now fully described in table methods. Regarding the cutoff point for the MRI signal, we defined diffusion‐weighted high signal intensity in the solid portion when signal in lesion is greater than urine, when b=1000 s/mm2. The following text was added to the methods section:

“We used a protocol aimed at assessing adnexal masses, which consisted of T2-weighted multiplanar sequences (axial, sagittal and coronal), a T1-weighted sequence in and out phase, diffusion-weighted sequence (b = 0, 500 and 1000) and T1-weighted sequences, with fat sat, before and after intravenous contrast injection with a power injection at a rate of 3.5mL/sec. The post-dynamic study consisted of 5 sequential acquisitions, with an interval of 30 seconds between them, with each sequence having an acquisition time varying between 10 and 13 seconds. The beginning of the first sequence was 21 seconds after the injection of intravenous contrast. An additional upper abdomen diffusion-weighted sequence was performed for screening of distant metastasis (solid organs or lymphadenopathy). 

We used an echo planar diffusion-weighted sequence (b = 0, 500 and 1000 s/mm2). The T2‐weighted signal intensity within the solid component of the adnexal mass 

is compared to urine within the bladder. A visual/qualitative analysis was performed: we defined diffusion‐weighted high signal intensity in the solid portion when signal in lesion is greater than urine, when b=1000 s/mm2. We emphasize that always in this analysis, we used the ADC (apparent diffusion coefficient) map as a reference, in order to avoid the effect T2 shines through (pseudo restriction).”

Comment 3: “Since histology is available for all the patients, an ROC analysis for the different imaging methods and combination with molecular biomarkers are suggested.”

Response 3: This is a valuable insight. However, due to an intrinsic and unsurmountable heterogeneity of histological diagnosis, a ROC analysis in this context would be not valid. For that reason, we opted to use standard thresholds for the markers, which would bring our analysis closer to the clinical practice.

---

## [Decision Letter · Decision Letter 1]

6 Mar 2023

Diffusion-weighted Magnetic Resonance sequence and CA125/CEA ratio can be used as add-on tools to ultrasound for the differentiation of ovarian from non-ovarian pelvic masses

PONE-D-22-12006R1

Dear Dr. Yoshida,

We’re pleased to inform you that your manuscript has been judged scientifically suitable for publication and will be formally accepted for publication once it meets all outstanding technical requirements.

Kind regards,

Federico Ferrari, MD, PhD

Academic Editor

PLOS ONE

Additional Editor Comments (optional):

Reviewers' comments:

Reviewer's Responses to Questions

**Comments to the Author**

1. If the authors have adequately addressed your comments raised in a previous round of review and you feel that this manuscript is now acceptable for publication, you may indicate that here to bypass the “Comments to the Author” section, enter your conflict of interest statement in the “Confidential to Editor” section, and submit your "Accept" recommendation.

Reviewer #1: All comments have been addressed

2. Is the manuscript technically sound, and do the data support the conclusions?

Reviewer #1: Yes

3. Has the statistical analysis been performed appropriately and rigorously? 

Reviewer #1: Yes

4. Have the authors made all data underlying the findings in their manuscript fully available?

Reviewer #1: Yes

5. Is the manuscript presented in an intelligible fashion and written in standard English?

Reviewer #1: Yes

6. Review Comments to the Author

Reviewer #1: All suggested corrections were made by the authors, and the manuscript can be accepted for publication.

7. PLOS authors have the option to publish the peer review history of their article (what does this mean?). If published, this will include your full peer review and any attached files.

Reviewer #1: No

---

## [Editor Report · Acceptance letter]

9 Mar 2023

PONE-D-22-12006R1 

Diffusion-weighted Magnetic Resonance sequence and CA125/CEA ratio can be used as add-on tools to ultrasound for the differentiation of ovarian from non-ovarian pelvic masses   

Dear Dr. Yoshida:

I'm pleased to inform you that your manuscript has been deemed suitable for publication in PLOS ONE. Congratulations! Your manuscript is now with our production department. 

Kind regards, 

on behalf of

Dr Federico Ferrari 

Academic Editor

PLOS ONE